# Investigation on the Flow Field Entropy Structure of Non-Synchronous Blade Vibration in an Axial Turbocompressor

**DOI:** 10.3390/e22121372

**Published:** 2020-12-04

**Authors:** Mingming Zhang, Anping Hou

**Affiliations:** 1Faculty of Science, Beijing University of Technology, Beijing 100124, China; mmzhang@bjut.edu.cn; 2School of Energy and Power, Beihang University, Beijing 100191, China

**Keywords:** axial turbocompressor, non-synchronous vibration, fluid-structure interaction, deviation of IGV, spiral vortex

## Abstract

In order to explore the inducing factors and mechanism of the non-synchronous vibration, the flow field structure and its formation mechanism in the non-synchronous vibration state of a high speed turbocompressor are discussed in this paper, based on the fluid–structure interaction method. The predicted frequencies *f_BV_* (4.4EO), *f_AR_* (9.6EO) in the field have a good correspondence with the experimental data, which verify the reliability and accuracy of the numerical method. The results indicate that, under a deviation in the adjustment of inlet guide vane (IGV), the disturbances of pressure in the tip diffuse upstream and downstream, and maintain the corresponding relationship with the non-synchronous vibration frequency of the blade. An instability flow that developed at the tip region of 90% span emerged due to interactions among the incoming main flow, the axial separation backflow, and the tip leakage vortices. The separation vortices in the blade passage mixed up with the tip leakage flow reverse at the trailing edge of blade tip, presenting a spiral vortex structure which flows upstream to the leading edge of the adjacent blade. The disturbances of the spiral vortexes emerge to rotate at 54.5% of the rotor speed in the same rotating direction as a modal oscillation. The blade vibration in the turbocompressor is found to be related to the unsteadiness of the tip flow. The large pressure oscillation caused by the movement of the spiral vortex is regarded as the one of the main drivers for the non-synchronous vibration for the present turbocompressor, besides the deviation in the adjustment of IGV.

## 1. Introduction

In order to improve the aerodynamic performance of the compressor, blades with high load and efficiency are widely adopted. The interaction between the unsteady flow and the blade structure is further strengthened, which makes the problem of blade vibration more and more prominent. As an atypical phenomenon of blade vibration, non-synchronous vibration (NSV) has attracted the attention of researchers in recent years. Non-synchronous vibration means that the blade vibration frequency is not synchronous with the rotor frequency. There is not an integer order relation between the blade vibration frequency and the rotor frequency.

Previous studies [1,2,3,4,5,6] show that the flow oscillation in the tip region is one of the drivers for non-synchronous vibrations, such as rotating instability. Baumgartner et al. [1] described the large amplitude vibration of the first bending of blades on a multistage high-pressure compressor. Its vibration frequency is not consistent with the rotor rotation frequency. It is shown that the NSV is associated with the fluid dynamic instability in the rotor tip region. The experiments conducted by Marz et al. [2] and Mailach et al. [3] indicate the occurrence of NSV with its frequency at a non-engine order of blade passing frequency. The rotating instability is located in the blade tip region, and a highly fluctuating vortex is captured at casing wall. To investigate the influence of tip clearance on rotating instability, a low-speed compressor experiment [3] was carried out. Measurements show that the rotating instability is located at the blade tip region, where a strong blade tip vortex is observed with a large tip clearance.

In the experiment [4], a high-speed axial compressor exhibits a non-engine order vibration of the first stage rotor blades which occurs before stall. Based on the Reynolds-averaged Navier–Strokes (RANS) method, Kielb et al. [4] conducted the simulation of the 1/7th annulus of the high-speed axial compressor rotor. It is shown that the NSV is caused by the vortex oscillation in the blade suction side and the tip flow instability near 75% span. Moreover, the frequency of blade vibration is locked continuously [5]. Sanders [6] analyzed the identification and prediction of NSV problem encountered in an aircraft engine. The experimental and time-accurate CFD methods are used to demonstrate that the flow induced NSV instability is initiated due to the stalling of the fan stator.

The work done by Vo [7] approaches the investigation as a study of the formation of short length-scale (spike) rotating stall disturbances. The computations are based on a time-accurate single blade passage of a subsonic axial compressor. The oscillatory flow was explained to be the driver for non-synchronous vibrations with tip flow instability. Subsequently, Thomassin et al. [8,9] conducted a series of research to investigate the role of tip clearance flow in the occurrence of NSV by experimental and numerical methods. A physical mechanism to explain the NSV phenomenon is proposed as the blade tip trailing edge flow acting likes an impinging jet on the pressure side. The enforced motion technique was applied by Spiker et al. [10] for non-synchronous vibration with an input motion of blade in a range of interested IBPAs. The limit cycle oscillation (LCO) is determined as the damping goes from positive to negative. Numerical investigations conducted by Drelet et al. [11,12] aim to determine the relationship between tip clearance, operating temperature, and the critical speed of the non-synchronous vibration. The non-synchronous vibration is interpreted as its resonance with tip leakage flow. The calculations carried by Hwang et al. [13] show that this kind of vibration is attributed to the fluctuating blade tip vortex. As the flow rate decreases, the flow instability develops due to the blockage of tip leakage flow.

Zha et al. [14,15,16,17,18,19] carried out a series of works with the same compressor of Kielb [2], established 1/7th and full annulus models for the 1.5-stage compressor. The RANS simulations [14] show that the tip clearance shape has a strong influence on the frequency of NSV. Considering the interaction of fluid and structure, the computations are conducted by deforming the blade motion as the given modes [15]. Later, the delayed detached eddy simulation is used to investigate the mechanism of non-synchronous vibration [17]. The results show the separation vortex of tornado structure is formed near the leading edge of the rotor blade tip propagating at the speed of a non-engine order frequency. Pay attention to the interaction of fluid and structure, the applications of the time-domain FSI method have been taken into practice to research related contents. The computations [18,19] are conducted to investigate the NSV by comparing the flow excitation with rigid blades and the blade vibration with fluid–structure interaction. It is indicated that the NSV of the compressor is caused by the tornado-like tip vortex in the rotor tip as rotating flow instability.

With the understanding of the non-synchronous vibration of the compressor blade, it is mostly considered that the aerodynamic flow instability caused by the leakage vortex in the tip region is accounted for by this phenomenon [20]. Blade mode damping is also taken into consideration [21]. Blade non-synchronous vibration in a transonic compressor was observed by Moller et al. [22]. By analyzing the development process of tip flow blockage, the oscillation of separated flow blockage at the rotor tip region is considered as an important factor for the aeroelastic stability of the compressor. The tip-timing measurements analysis for non-synchronous vibration are presented in [23,24,25], and provide information for understanding the blade vibrating behaviors. This NSV phenomenon become a major interest for acoustic investigations yet, with establishing an analytical sound model [26]. The non-synchronous oscillations are identified by an acoustic mode, showing a dominant propagating acoustic mode interacting with vibrations of the rotor [26,27]. An explanation for this is presented for the lock-in between aerodynamic disturbances and the vibration pattern during NSV [28]. A semi-analytical model based on single degree of freedom oscillators is developed, which can reproduce the results from the reduced-domain computations.

Note that the induction factors of the non-synchronous blade vibration of compressor are complex, which reflect the coupling effects of unsteady flows and blade structure. To describe a new NSV of a high speed turbocompressor encountered, a time domain numerical approach is carried out by the fluid-structure iterative coupling method under the parallel computation. The aim of this work is to capture the dominant blade frequencies of turbocompressor, explore the induction factors, and investigate the flow entropy structure and mechanism in the non-synchronous vibration. The result indicates that it is sufficient to predict the NSV frequencies by capturing the resonance of flow instability and blade vibration with considering fluid-structure interaction. A flow-induced vibration is initiated by the large amplitude oscillating leakage flow in the tip regions, which is regarded as the driver for the NSV. This paper is organized as follows. Detailed descriptions of the numerical method and model are given in Section 2. Then, the verification of the simulations is performed with a comparison of experimental data in Section 3. Thereafter, in Section 4, the fluid mechanics of the results are discussed with the analysis of tip clearance flow and the blade resonance.

## 2. Descriptions of Numerical Approach

The 1.5 stage of a multistage axial high speed turbocompressor is chosen here as the research object in this paper. The test rig consists of 6 struts, 42 inlet guide vanes, 38 rotor blades and 82 stator blades. In the experiment, the turbocompressor exhibits a large amplitude blade vibration in a certain speed range (9700 rpm~10,190 rpm), which is a non-engine order of the rotor. This kind of vibration is observed to be in a frequency-locked state, which is close to the blade first bending mode. Moreover, it is found that there exists a 2°deviation in the adjustment of the adjustable guide vane, which leads to an effect of mistuning on the installation angles of guide vanes, as shown in Figure 1.

According to the number of blades in each row and the mistuning distribution of the guide vanes in the 1.5 stage high speed turbocompressor, the research object is simplified to a half annulus configuration model. As the non-synchronous vibration phenomenon of the turbocompressor involves the coupling of unsteady vortex flow and blade vibration, this paper adopts the time-domain fluid structure coupling iterative numerical simulation method [29,30] to reveal the detailed mechanism of NSV. The commercial CFD software ANSYS Package is used for the current calculation. The unsteady flow field in the turbocompressor is solved by using the numerical solution of 3-D Navier-Stokes equations adopted by the commercial software CFX. The spatial discretization of the flow governing equations is employed on an upwind scheme, and second-order backward differencing is integrated for the time-accurate solution [31]. Boundary conditions imposed on the inlet consist of total pressure and total temperature. The incoming flow direction is set normally to the inlet guide vanes with turbulence intensity for 5%. A specified average static pressure is implemented at the exit boundary with a constant high rotor speed. Smooth, adiabatic, and no-slip wall boundary conditions are applied for the flow field solution. The numerical model of the 1.5-stage turbocompressor is shown in Figure 2.

For the consideration of computation cost, an attempt is made to reach a balance between the computation accuracy and efficiency. Because of the importance of the boundary separations and tip leakage vortices, the mesh refinement in these regions is made. The independence of the calculation results to the mesh scale is conducted by a range of grid sizes through steady computations. The standard *k-ε* function is used as the turbulence model for the wall treatment by dealing with a coarse grid. Y-plus of the meshes near the blade surface is kept smaller than 10 in all cases. Finally, the quantitative mesh elements are approximately selected as 8.09 million based on performances under different grid densities (Figure 3), still kept relatively coarse to limit the computational cost. The distributions of the cells in the used mesh are provided in the Table 1 with the circumferential, axial, radial direction resolution. The finite element model of the structure is shown in Figure 4 with all the blades considered. The dynamic frequency of the first order bending mode is calculated to be 751 Hz at the current rotational speed under 10,190 rpm. During the unsteady computations, every rotor blade pitch is divided into 10 time steps, with each time-step including a maximum of 20 inner interaction steps. Then the time step is calculated to be 1.5495 × 10^−5^ s. Convergent numerical results are obtained at least after five revolutions.

## 3. Numerical Reconstruction of the Non-Synchronous Vibration

Regarding the non-synchronous vibration of a high speed turbocompressor, a reproduction for this vibration phenomenon is carried out numerically by the time domain fluid–structure iterative coupling computation. The reliability and accuracy of the numerical method are verified by compared with the data of experiment. Investigations of the flow entropy structure and the dominant blade frequencies captured provide a theoretical support for the mechanism research in the non-synchronous vibration.

It was found in the experiment that vibration of the rotor blade in first stage is locked as the first order bending mode in a certain rotational speed range under a deviation in the adjustment of IGV. A blade vibration with high amplitude appears as frequency-locked and phase-locked, which exhibits a non-integral order (4.4~4.6) relationship with the rotating speed. Yang [32] analyzed the experiment results of the surface stress of the rotor blade and the pressure pulsation on the casing wall, and compared with the acoustic resonance phenomenon. It is indicated that there may be an acoustic resonance phenomenon in the operation state of the turbocompressor. The influence of sound field on the non-synchronous vibration of the rotor blade is studied by Han [33]. The analysis on the turbocompressor noise which is monitored at the casing wall of the turbocompressor shows that the characteristic frequency of the noise signal appears simultaneously with the occurrence of non-synchronous vibration of the blade. The turbocompressor rotating speed and adjustment of guide vane are the main factors affecting the blade vibration of the turbocompressor.

According to the dynamic pressure data at 90% span of the blade surface, the spectrum results in the experiment [32] show that the dominant frequency of pressure fluctuation at the blade surface is consistent with the first bending vibration frequency of blade when the stress of blade correspondingly increases. Under the current operating speed of 10190 rpm, the fluid–structure coupled computation is performed by solving the unsteady Navier–Stokes equations, coupled with the modal structural equations. Figure 5 plots the displacement deformation of the turbocompressor blade in this operating state. The numerical results indicate that the vibration of the blade exhibits the first bending mode during the process of vibration. Moreover, the maximum displacement deformation point appears at the trailing edge of the blade tip.

The numerical monitor is attached at the node of maximum deformation on the blade tip trailing edge to observe the displacement variation during the vibration. As indicated in the Figure 6, the blade vibration obviously possesses the characteristic of a beat wave. To inspect the NSV with frequency spectrum, the analysis of the blade vibration displacement is sketched in Figure 7. Due to the local mistuning installation angles of the inlet guide vanes, the two times rotating frequency and its multiplied frequencies are captured clearly. Besides these frequencies in multiples of the engine order (EO), it can be seen that the frequency *f_BV_* (blade vibration) of 747 Hz is in the dominant status of blade vibration, which is around 4.4 times of the engine order (EO). This is consistent with the range of 4.4~4.6EO in the experiment, which is close to the blade first bending mode. This means that the non-synchronous vibration of the rotor blade appears in the form of the first bending mode.

Besides the analysis of the deformation of the blade, the unsteady pressure captured at the casing wall is used to observe the content of frequency, as shown in Figure 8. It is found that the dominant frequencies of pressure fluctuation appear as the characteristic frequency *f_AR_* (acoustic resonance) and the frequency modulation of *f_AR_* and *f_BPF_* (blade passing frequency). The characteristic frequency *f_AR_* is a non-integral order of the rotor frequency, and the occurrence of this characteristic frequency is in good synchronization with the appearance of the first order bending vibration of the rotor blade. The relationship between *f_AR_* and *f_BV_* will be discussed in the following part.

The comparisons between the simulation results and measurements are listed in the Table 2. For each validation variable it is pointed out whether the result is within in the deviation range or not. The prediction of the vibration frequency by computation is in reasonable agreement with the experimental data, as well as the IBPA. However, because of the limitation of CFD in simulating the turbulence, the characteristic frequency *f_AR_* is exceeding the range of measurement. Despite this inaccuracy, the current study demonstrates that it is sufficient to predict the NSV phenomenon by capturing the resonance of flow instability and blade vibration in consideration of the fluid–structure interaction. The analysis of the flow entropy structure and mechanism in the non-synchronous vibration was performed to explore the induction factors in detail thereafter.

## 4. Investigation on the Flow Field Structure in Non-Synchronous Vibration

From the analysis above, it can be seen that the non-synchronous vibration phenomenon of the rotor blade in a high speed turbocompressor can be accurately reproduced in this paper through the time-domain fluid structure coupling iterative numerical simulation method. The calculated NSV frequency, flow field characteristic frequencies, and other parameters have a good corresponding relationship with the experimental data. Moreover, the results calculated provide a basis for the study of the induction factors of blade non-synchronous vibration and its fluid–structure coupling mechanism.

### 4.1. The Origin of Disturbance in the Turbocompressor

In order to detect the flow entropy structure of turbulence in a flow field and the characteristic frequency, a number of numerical sensors are set up in different sections to acquire the pressure signals. With four numerical probes located in the middle of tip clearance, four specific sections arranged at interfaces between the rotor row and the stator row are mounted at axial and radial directions of the model, which are displayed in Figure 9.

As shown in the Mach number contour (Figure 10) near the rotor leading edge (2-2 section), the unsteadiness of the flow field has been found to be inherent. The passages above 85% span of the blade are blocked by the low speed fluid. The unsteady vortices especially in the tip region with stall separation have a strong influence on the interaction of fluid structure coupling with the blade. Since the unstable flow in the field is mainly concentrated in the area around 90% span of the blade of the turbocompressor, the static pressure signals in different monitoring sections at interfaces are acquired to inspect the mechanism in the NSV.

The propagation characteristic of vortex and the source of the flow separation are examined by the analysis on spectrum. The results indicate that the dominant frequency *f_AR_* (1630 Hz) of the pressure fluctuation appears accordantly in every monitoring section, which is located to be 9.6EO, as shown in Figure 11. This characteristic frequency *f_AR_* reflects the non-integral order of the rotor frequency, and it spreads out from the rotor domain as the source to upstream and downstream with the strength weakening. It is shown the fluctuations of pressure around the rotor are larger than those at other monitoring sections. The peak amplitude is found to be near the rotor leading edge where the maximum pressure oscillation occurs due to the center of the separation vortex.

### 4.2. The Relationship of the Frequencies in the Field

Note that the pressure signal acquired at the rotor leading edge is shown to be in a frequency-locked manner, which is accordant to the unsteady pressure captured at the casing wall. Moreover, the fluctuation of aerodynamic force is inevitably induced by this high amplitude of unsteady pressure excitation during the vibration of blade. However, what kind of physical information is represented by the characteristic frequency *f_AR_*? What is the relationship between the characteristic frequency *f_AR_* and the vibration frequency *f_BV_* of the blade? In order to answer these questions, the propagation characteristic from the separation vortex core to the spatial field will be studied further. Further, the relationship between the separation flow as the inducing factor and the blade vibration is thoroughly discussed as well.

In the process of blade vibration, there exists a phase angle between adjacent blades within a certain angle range, which is called inter blade phase angle (IBPA). In the present study, the blade phase angles calculated during the vibration have been visualized circularly as a sinusoidal like wave superimposed on the rotor in Figure 12. Each blade deflects from a certain angle in sequence approximately. The current results shown up represent a family of spatial harmonics composed by the superposition of a number of rotating nodal diameter patterns. The vibration of each blade is affected by the nodal diameter with different phase indexing. The calculated average value of the inter blade phase angle is 132.6°. According to the equation below:(1)$$D=\frac{\alpha_{ibpa}}{2\pi }\times N$$
where *D* represents the nodal diameter number, $\alpha_{ibpa}$ expresses the inter blade phase angle, *N* means the number of blades. The nodal diameter in this computation is computed to be 14, which indicates that there is a pressure traveling wave with a pitch diameter number of 14 propagates along the circumferential direction.

The characteristic frequency *f_BV_* of the rotor blade in the rotating coordinate system is 4.4 EO, while the corresponding characteristic frequency *f_AR_* in the flow filed of the stationary coordinate system is 9.6 EO. The sum of *f_BV_* and *f_AR_* is just equal to the nodal diameter number. This means that the non-synchronous frequency *f_BV_* of the blade vibration is the representation of the characteristic frequency *f_AR_* in the stationary coordinate system at each measurement section. That is to say, the performance of the flow field and the estimated vibration of blade in the turbocompressor are found to be related to the unsteadiness of the tip flow at blade leading edge. The predicted dominant frequency *f_AR_* is not harmonic to the engine order. The non-synchronous vibration of the blade is induced due to the interactions among the unsteady tip flow and the blade structure. The instability of vortices in the vicinity of the rotor tip is considered to be the main cause of the excitation of blade vibration.

### 4.3. The Flow Entropy Structure of Non-Synchronous Vibration

From the above analysis, it can be seen that the source of unstable flow in the high speed turbocompressor is concentrated in the rotor domain, and the induced factors of the non-synchronous vibration of the blade are generated in the tip region. Therefore, in order to study the correlation of unsteady leakage flow and blade vibration, the static pressure signals are captured by the probes located in the middle of tip clearance at equal intervals in the chordwise direction, as the location shown in Figure 10. Then, these signals are extracted to observe the information behind the vortex of the tip leakage flow in the rotating coordinate system.

As sketched in Figure 13, the obtained spectrum information at tip region is abundant, which is indicated that the unsteadiness is noticeable in the leakage flow. The pressure oscillations with the dominant frequency of 747 Hz (4.4EO) emerge before 60% chord of the blade in the tip region. This dominant frequency is consistent with the vibration frequency of the first bending mode of the blade. However, the amplitude of the frequency 747 Hz decrease rapidly afterwards. Instead, the pressure oscillations with the frequency of 4EO are found to be prominent at trailing edge of the blade tip. Due to the local mistuning installation angles of the inlet guide vanes, the frequencies in multiples of 2EO are viewed to be accompanied all along the tip, as the response to the wake of vanes upstream. Also, it is revealed that the vortexes associated with the non-synchronous vibration are assembled at the first half of the blade chord. So, it is necessary to analyze the flow structure at the leading edge of blade tip.

Based on the computation results at a specified operating point, instantaneous distributions of the local flow are marked to understand the unsteady flow structure. In the velocity vector diagram on the 90% span of the blade, an inverse flow behavior exists in the passage. Figure 14 shows that the separation vortexes in the blade passages occupy the main part of flow in the blade tip region. Affected by the separation flow blocked at the suction side, the main flow could not continue to flow along the axial direction smoothly after entering into the flow path. Moreover, the extent of flow blockage and loss increases sharply in the suction side of the blade at the tip trailing edge. Then, the main flow mixed with the separation vortexes in the tip area has been divided into two streams, and one of the smaller parts of the flow keeps running downstream.

Because of the blockage at near the blade suction side, a countercurrent phenomenon occurs and a part of flow in blade passage is traversed at the rotor outlet. The reversal flow is clearly identified in Figure 14 as drawn by a red line at almost 80% of blade to blade section near the tip region, which gives rise to a significant increase in entropy. With the movement of the separation flow, the blockage at tip region is pushed forward. In the circumferential direction, the interface of the main flow and the separation vortex in the passage is in coincidence with the leading edge of the adjacent blades at inlet. It can be concluded that the unsteadiness of the separation flow plays a key role on such a backflow performance. The regions that have a negative axial velocity composed by the separation vortex are noticed to exist near the casing boundary. The blockage near the casing is accumulated due to the separation vortex mixed with tip leakage flow spreading out circumferentially. The interaction of the upstream main flow and the local separation flow is responsible for the periodical flows in the passages.

In order to observe whether the same phenomenon exists in other states, some more computations are conducted by changing the static pressure implemented at the exit. The simulated state points are shown in Figure 15. According to the spectrum diagram of blade displacement in state B, sketched in Figure 16, it can be seen that the frequencies of blade vibration are in multiples of the engine order. There is not a non-synchronous vibration, which is different from the state C (NSV point). When the turbocompressor operating away from the state C, the flow obviously enters the blade passages along the axial direction, and follows the flow path smoothly to the exit near the blade trailing edge, as shown in Figure 17. The reversal flow doesn’t happen in this operating condition.

However, at the fault status of NSV (state C), the helical vortex structure is observed at the leading edge in the blade tip, as illustrated from the perspective of three-dimension in the Figure 18. Unlike the regular tip separation vortices which are out of order, the observed vortex in this study swirls strongly at the blade suction surface in the tip region. After entering into the passage the axis of the vortex rotation is perpendicular to the suction surface of the blade practically. It can be seen that, affected by the leakage flow, the vortices in passage reverse at the trailing edge of the blade tip. Then, the separation vortices of the blade suction surface present a spiral vortex structure, and flow upstream to the leading edge of the adjacent blade. Mixed up with the main flow of the adjacent passage, the vortices continue to form another unstable vortex with a spiral structure.

It can be seen from the Figure 19 that the spiral vortex located in the leading tip rotates with the blades, but at a lower speed. Moreover, the high entropy regions appear alternately. The entropy of flow field is depicted in time-sequences to show the periodic evolution of the flow structure in the leading edge of the rotor tip. In a cycle of non-synchronous vibration, the strength of the vortex experienced by the blade tip is constantly changing with time. At the time of 0/2 *T_nsv_*, there is an entropy region formed by vortices in the leading edge of the monitor blade, which presents a bit weaker strength than that of the adjacent passage. As approaching to time of 1/2 *T_nsv_* the rotating blade catches up the vortex which is seated in the adjacent blade tip at former moment. Further, the amplitude of aerodynamic disturbance on the blade reaches the maximum. The blade experiences a process of gradual increase of vortex intensity. Then, in the next period of 1/2 *T_nsv_*, the blade continues to scan another adjacent weak spiral vortex, with vortex intensity felt by the blade restoring to the initial state of the cycle. The disturbances of the spiral vortexes propagate at 54.5% of the rotor speed, indicating a pattern as modal oscillation. In the process of rotor rotation, the blade continuously pursues the vortex in front. When the frequency of the blade sweeping vortex coincides with the natural mode of the blade, the blade can demonstrate a large amplitude vibration. Moreovere, the frequency to sweep the vortexes by the blade becomes the frequency of blade asynchronous vibration.

The results show that the complex spiral vortex composed of the separation vortex shedding and tip leakage flow is mainly concentrated in the area near the blade suction side. The spiral vortex situated near the blade tip travels in the circumferential direction at a blade span of about 85% and forms a vortex ring around the annulus. Originating from the interaction of the separation flow and the tip leakage vortex, the mixed vortices induce the reversal flow at the blade tip region. Due to the flow coming and leaving, the oscillating phenomenon of the flow leads to a large fluctuation of the local pressure. This unstable flow existing in the tip domain results in another oscillation of aerodynamic force on the blade, which urges the blade to vibrate in mode. This shows that the blade vibration in the present investigation is a typical flow induced vibration coupled to the tip flow instability.

## 5. Discussions

It is discussed that an instability flow that developed at the tip region emerged due to interactions among the incoming main flow, the axial backflow, and the tip leakage vortices. The presence of the separation backflow at the blade tip trailing edge could impinge on the pressure side of the adjacent blade. This kind of rotating instability emerged with the spiral vortex, and the vortex core shows a clear presence in every rotor passage near 60% chord length of the blade tip. This phenomenon seems similar with the rotating stall. However, there still exists a little difference. As depicted in literature [31], near the stall boundary, the interface of the incoming flow and tip leakage vortex is pushed to the leading edge of the neighboring blade. The increasing blockage at tip region is concentrated on the local suction side of the blade, rotating with the blade. However, the spiral vortex presented has its own propagation speed in the circumferential direction, possesses a relative speed with the blade. Moreover, there is not a sharp decrease in the performance of the stage, which is obviously different from the rotating stall.

It can be discerned that the flow oscillations related to the instability analyzed above are the cause of the impingement on the pressure side just below the blade tip. When these separation vortexes become powerful enough to alter the blade loading of the neighboring blade, the blade vibration with a large amplitude will be stimulated soon. The spiral vortex observed in the flow instability of blade tip is considered as the one of the main causes to the non-synchronous vibration for the present turbocompressor. However, the deviation in the adjustment of IGV is the fundamental cause for the flow performance of the stage. Under the normal adjustment of IGV in the experiment, the stress of rotor blade in the first stage doesn’t present a sudden increase appearance among the speed range. The research on non-synchronous vibration caused by the regulation of IGV is considered to be a novelty in this paper, which is rarely addressed in the previous literatures. The influence of guide vane regulation on the blade vibration should be studied in depth, especially for the non-synchronous vibration. The characteristics of blade vibration are worthy of further analysis during the initiation and development of non-synchronous vibration at different operating states.

For a multistage turbocompressor, there are many differences in the flow structure of each stage. Indeed, the non-synchronous blade vibration is observed to appear in the first stage due to a deviation of 2° in the adjustment of the adjustable guide vane in the experiment. The effect of the deviation of IGV is gradually weakened downstream in the multistage turbocompressor. Generally speaking, the flow separation of the high-pressure stages is much more serious because of the reverse pressure gradient. So the instability of rotating stall may be more likely evolved instead of the spiral vortex appearance described above. So, whether the conclusions for the cause of NSV are applicable depends on the actual situation of each stage in the multistage turbocompressor.

## 6. Conclusions

In order to explore the inducing factors and mechanism of the non-synchronous vibration, a time domain numerical approach was followed using the fluid–structure iterative coupling method under the parallel computation. Through a comparison with experimental data, the reliability and accuracy of the numerical method are verified. For the study of the induction factors and the fluid–structure coupling mechanism, the flow entropy structure of blade non-synchronous vibration is discussed in detail. The main conclusions can be summarized as follows:

(1) This paper numerically reproduces the non-synchronous vibration phenomenon of a high speed turbocompressor under a deviation in the adjustment of the inlet guide vane. The calculated characteristic frequency *f_AR_* (9.6EO) is a non-integral order of the rotor frequency, and the occurrence of this characteristic frequency is in a good synchronization with appearance of the first order bending vibration of the rotor blade. The predicted frequencies have a good correspondence with the experimental data, which verifies the reliability and accuracy of the numerical method.

(2) The characteristic frequency *f_AR_* reflects the non-integral order of the rotor frequency, and it spreads out from the rotor domain as the source to upstream and downstream with the strength weakening. The non-synchronous frequency *f_BV_* (4.4EO) of the blade vibration is the representation of the characteristic frequency *f_AR_* in the stationary coordinate system. The sum of *f_BV_* and *f_AR_* is just equal to the nodal diameter number 14. The performance of the flow field and the estimated vibration of blade in the turbocompressor are found to be related to the unsteadiness of the tip flow at blade leading edge of 90% span.

(3) An instability flow that developed at the tip region emerged due to interactions among the incoming main flow, the axial backflow, and the tip leakage vortices. Affected by the leakage flow, the vortices in passage reverse at the trailing edge of blade tip, and the separation vortices of the blade suction surface present a spiral vortex structure flowing upstream to the leading edge of the adjacent blade. The disturbances of the spiral vortexes emerge to rotate at 54.5% of the rotor speed in the same rotating direction as a modal wave. Due to the flow coming and leaving, the oscillating flow leads to a large fluctuation of the local pressure, causing an oscillation of aerodynamic force on the blade. A flow induced vibration is initiated by the large amplitude oscillating spiral vortex in the tip region, which is regarded as the one of the main drivers for the non-synchronous vibration for the present turbocompressor.

(4) This kind of rotating instability demonstrated by the spiral vortex is observed to be different from the local separation flow and rotating stall. The spiral vortex propagates at a relative speed of the blade in the circumferential direction with no performance degradation of the stage. The rotating spiral vortex is investigated as the one of the main factors affecting the non-synchronous vibration, but the deviation in the adjustment of IGV is considered to be the fundamental cause. So, the influence of guide vane regulation on the blade vibration is worthy of further analysis, especially for the non-synchronous vibration.

## Figures and Tables

**Figure 1 entropy-22-01372-f001:**
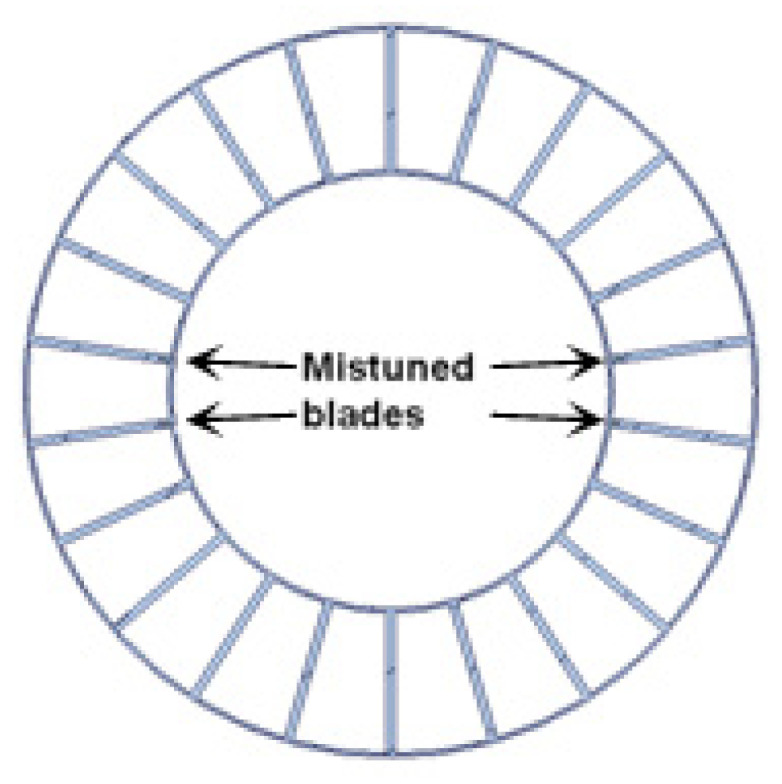
The mistuning distribution of the guide vanes.

**Figure 2 entropy-22-01372-f002:**
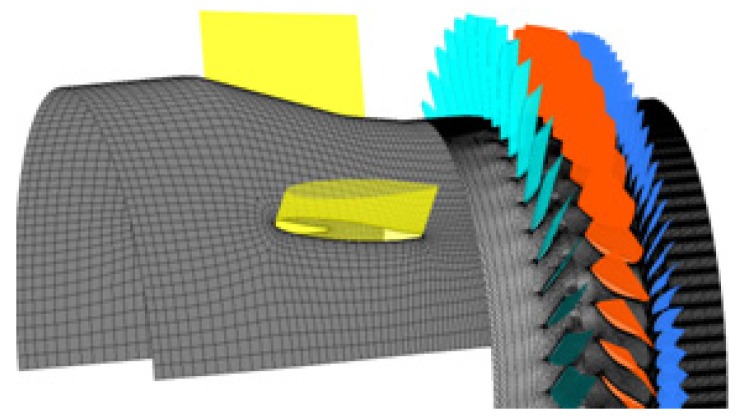
Numerical model of 1.5 stage high speed turbocompressor.

**Figure 3 entropy-22-01372-f003:**
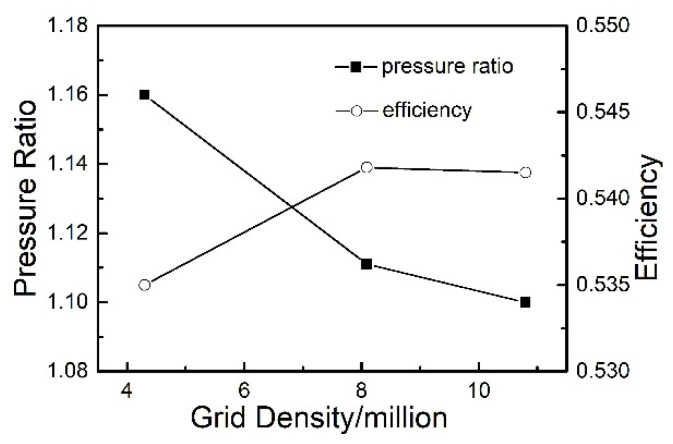
Performances under grid density.

**Figure 4 entropy-22-01372-f004:**
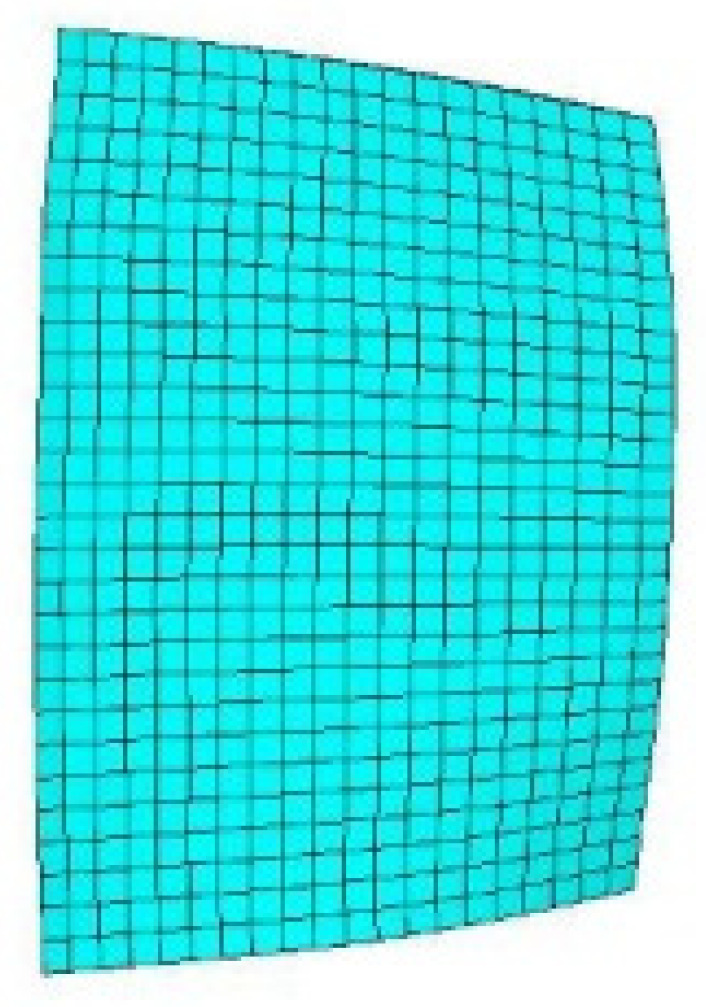
Structural grid of the rotor blades.

**Figure 5 entropy-22-01372-f005:**
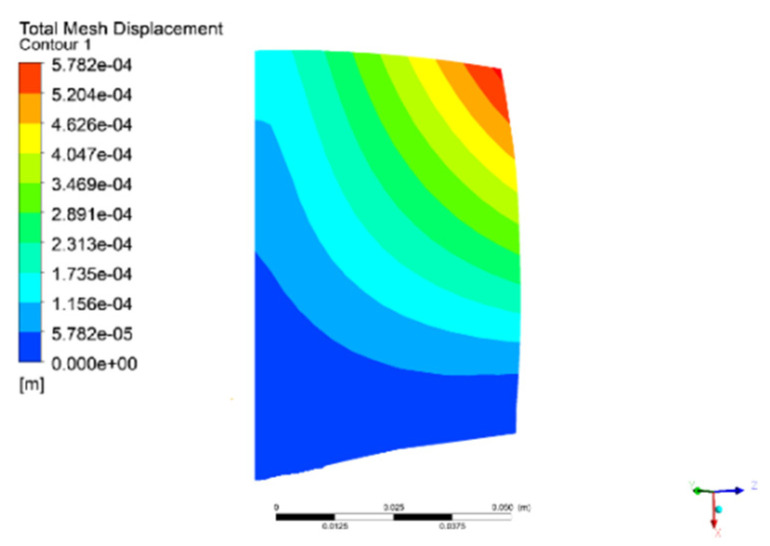
Deformation contour of blade vibration.

**Figure 6 entropy-22-01372-f006:**
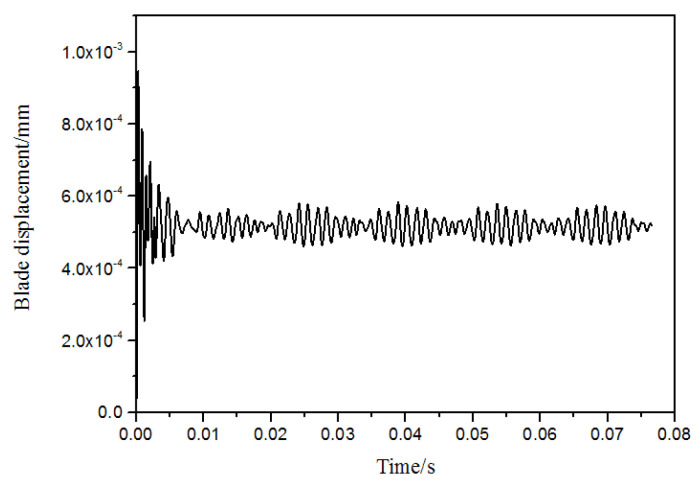
Time-domain curve of blade vibration displacement.

**Figure 7 entropy-22-01372-f007:**
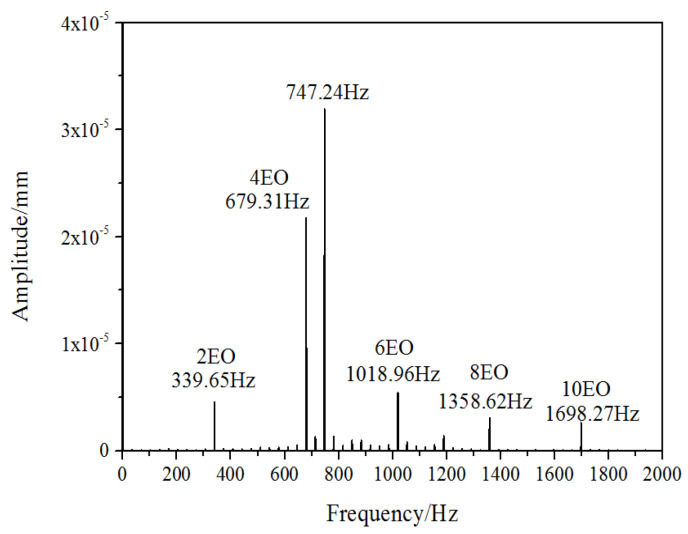
Spectrum diagram of blade vibration displacement.

**Figure 8 entropy-22-01372-f008:**
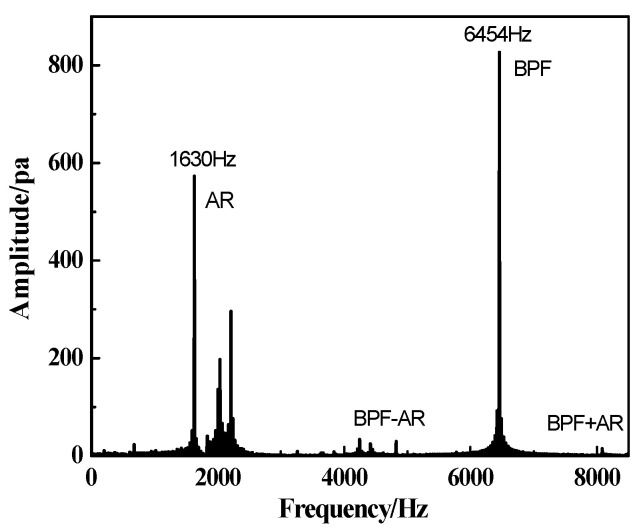
Spectrum analysis of unsteady pressure at casing wall.

**Figure 9 entropy-22-01372-f009:**
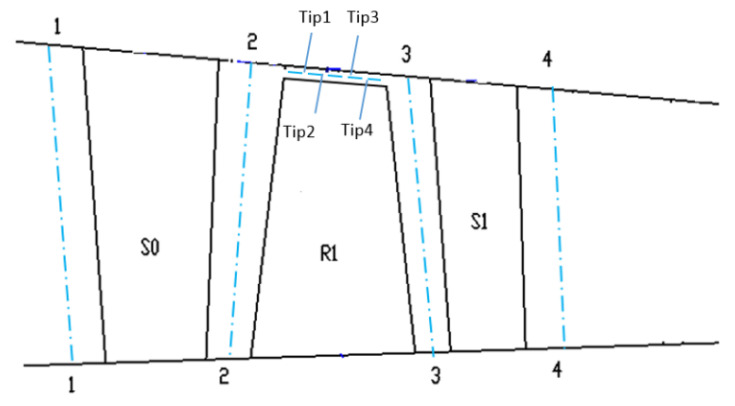
Measurement sections in the turbocompressor model.

**Figure 10 entropy-22-01372-f010:**
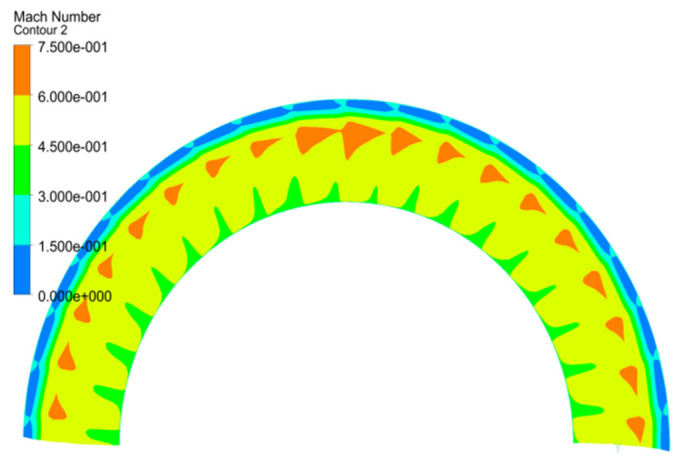
Mach number contour of the inlet section.

**Figure 11 entropy-22-01372-f011:**
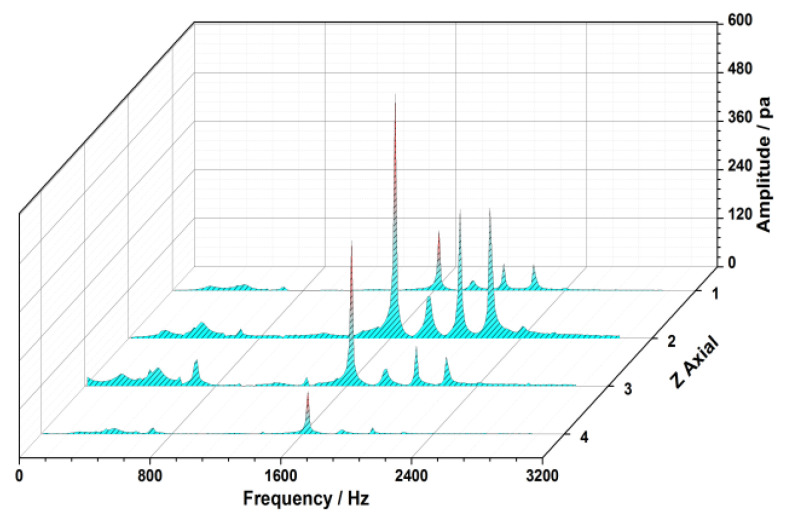
Axial propagation characteristic of unsteady vortex flow.

**Figure 12 entropy-22-01372-f012:**
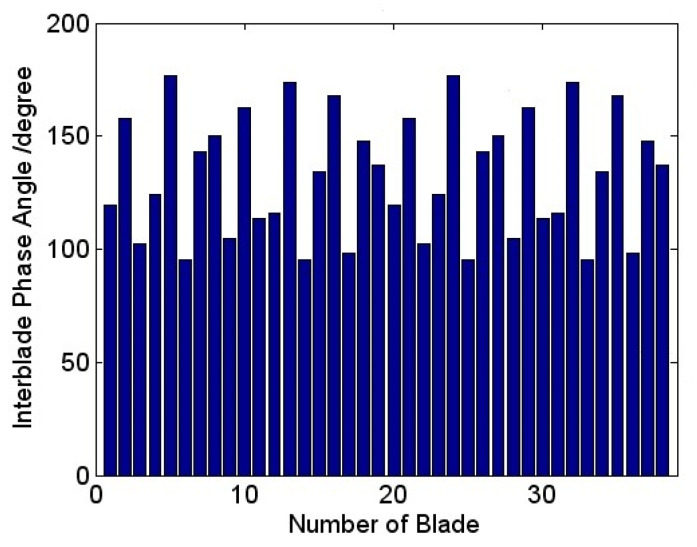
Blade vibration phase angle.

**Figure 13 entropy-22-01372-f013:**
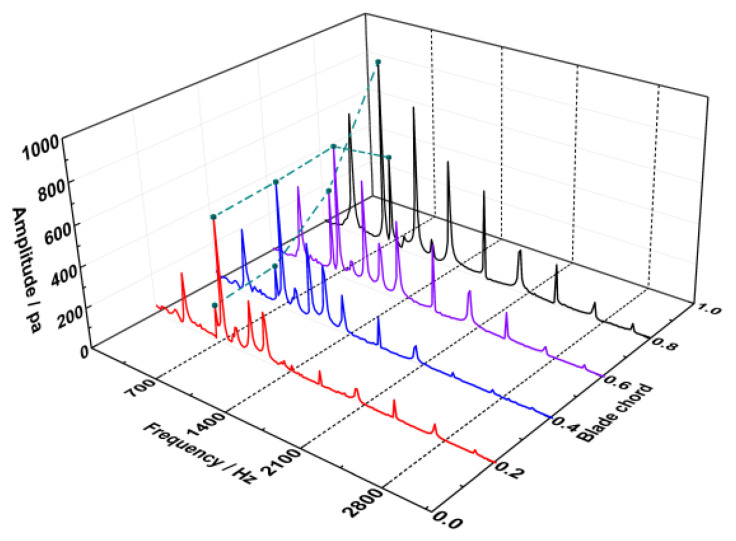
Spectrum on the vortex in the tip clearance.

**Figure 14 entropy-22-01372-f014:**
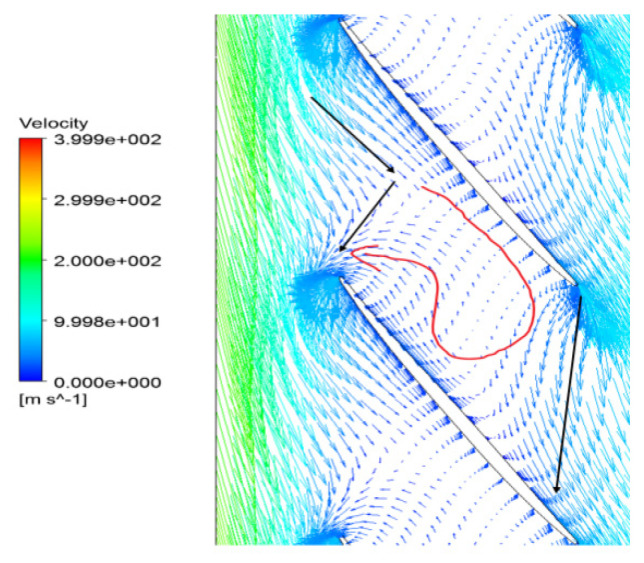
Velocity vector graph at the 90% span of blade.

**Figure 15 entropy-22-01372-f015:**
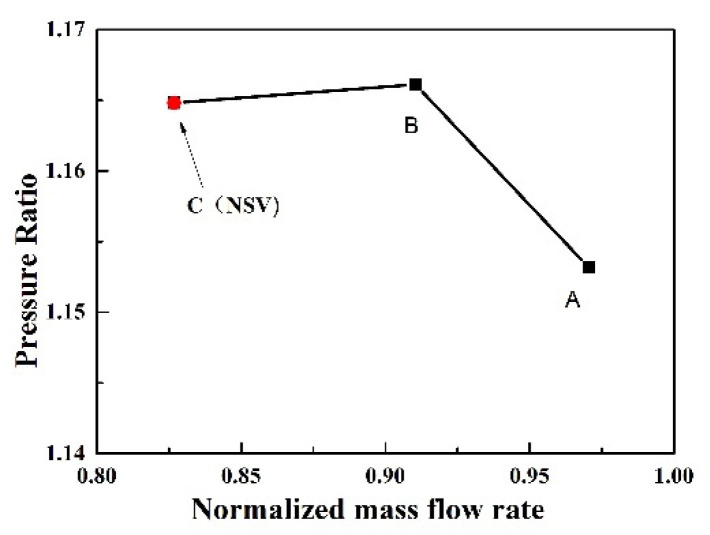
Operating States in computation.

**Figure 16 entropy-22-01372-f016:**
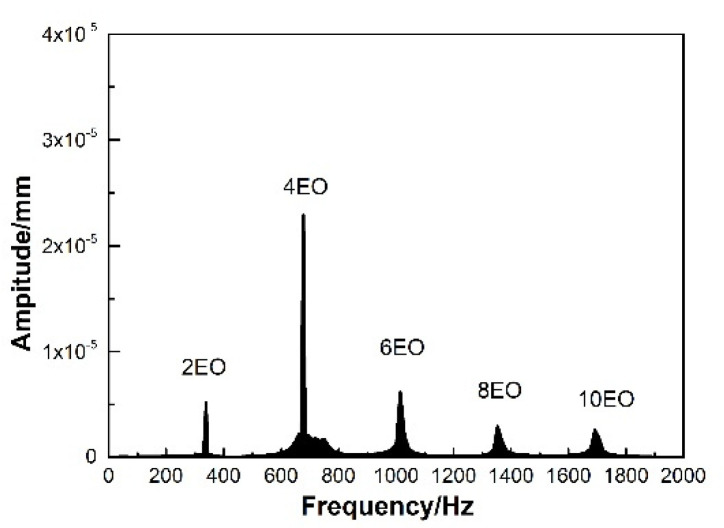
Spectrum diagram of blade vibration displacement at state B.

**Figure 17 entropy-22-01372-f017:**
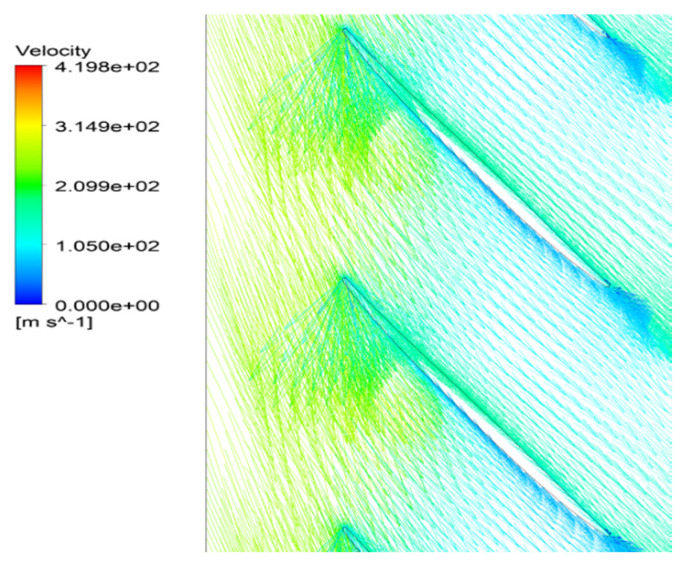
Velocity vector graph at the 95% span of blade at state B.

**Figure 18 entropy-22-01372-f018:**
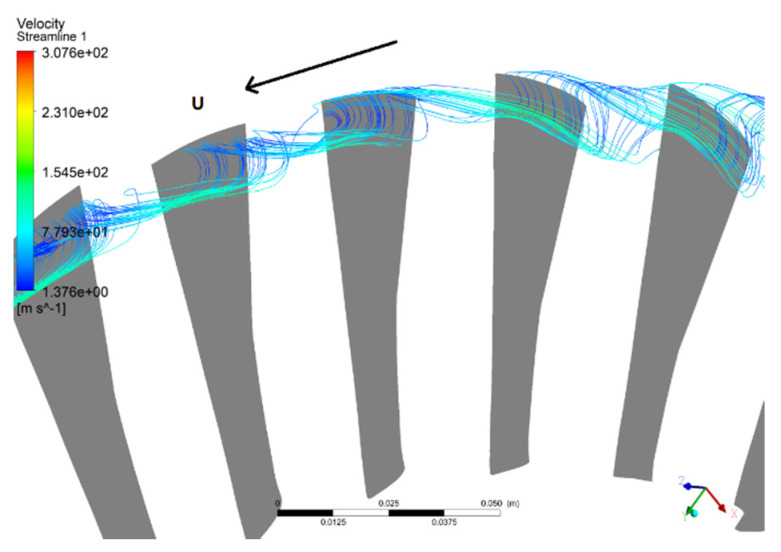
Spiral vortex at the leading edge in the blade tip.

**Figure 19 entropy-22-01372-f019:**
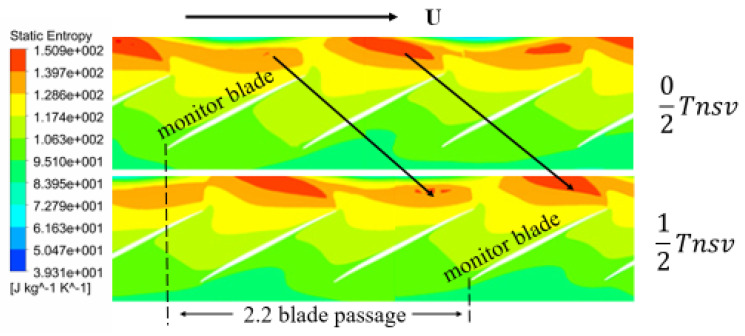
Entropy of the spiral vortex at the 90% span of blade.

**Table 1 entropy-22-01372-t001:** Distributions of grids.

Zone	Domain	Size/Domain	Size/Zone
Strut	3	73 × 91 × 47	93 × 10^4^
IGV	21	37 × 45 × 57	199 × 10^4^
Rotor	19	37 × 77 × 67	362 × 10^4^
entry 2	41	19 × 35 × 57	155 × 10^4^
Total			809 × 10^4^

**Table 2 entropy-22-01372-t002:** Distributions of grids.

Validation Variable	Vibration Mode	*F_BV_*/EO	*F_AR_*/EO	IBPA
measurement	first flexure	4.4~4.6	8~9	113°~133°
computation	first flexure	4.4	9.6	132.6°
deviation range	√	√	×	√

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
