# Peer review of "Investigation on the Flow Field Entropy Structure of Non-Synchronous Blade Vibration in an Axial Turbocompressor"

_entropy, 2020, doi:10.3390/e22121372_

Round 1

Reviewer 1 Report

Summary:

In this work the authors present the results of a numerical investigation of Non-Synchronous-Vibration (NSV) in the context of an axial compressor rig based on time-accurate URANS simulations using the commercial CFD code ANSYS. Following a detailed, and somewhat lengthy, review of the current literature a brief description of the numerical model and compressor configuration are presented before the authors then provide a detailed analysis and description of the complex flow physics. Overall, the authors address a topic of significant current interest to the turbomachinery community and provide insight into the underlying flow physics of NSV. However, in its current form the article is difficult to read and therefore needs to be significantly revised before it should be considered for publication. Furthermore, I am skeptical about the appropriateness of the article for current journal.

General Remarks:

Foremost, the English needs to be significantly improved as currently this impedes the reader’s ability to follow and understand the authors’ description of the complex flow physics presented in the article. Within this context I would strongly recommend that the authors include additional images of the flow, possibly in the form of time sequences, to both support their descriptions of the flow physics and aid the reader in understanding their observations. I would also like to see some more results, such as a comparison of the experimental and numerical operating points, to confirm the validity of the numerical model. What do the authors mean when referring to the flow field entropy structures? What is an entropy structure?

Specific Corrections:

  • Descriptions of numerical approach: Include a more detailed description of the compressor rig. E.g. Blade counts, RPM, operating point.
  • Descriptions of numerical approach: Alongside the total number of cells used in the mesh please provide details of their distribution across the blade rows and state the radial resolution. How many time-steps were used to resolve the flow unsteadiness? What y+ values does the mesh. Is the mesh low Reynolds or are wall functions being used? Why was the strut included in the numerical model? Was this important?
  • Descriptions of numerical approach, Figure 3: I don’t see the necessity of this figure and suggest it be dropped.
  • Descriptions of numerical approach, Figure 4: As all blades are meshed identically, it should be sufficient to simply display the mesh of a single blade. Although, I think this figure could be dropped as it has no real direct relevance to the study.
  • Numerical reconstruction of…, Figure 5: Again, it should be sufficient to show the displacement (eigenmode) of a single blade.
  • The Flow Entropy of Structure of NSV: To aid understanding the complex flow physics I strongly suggest including time-sequences to help the reader better understand your observation. Presumably, this would also make it easier to describe the observed phenomena.

Author Response

Reply to Reviewer Paper: entropy-1006425

Title: Investigation on the Flow Field Entropy Structure of Non-Synchronous Blade Vibration in an Axial Compressor

Thank you very much for the suggestions for our draft. We have replied point by point to your comments, along with a clear indication of red color at the location of the revised paper. The comments and replies can be summarized as follows: 

Q: Within this context I would strongly recommend that the authors include additional images of the flow, possibly in the form of time sequences, to both support their descriptions of the flow physics and aid the reader in understanding their observations. I would also like to see some more results, such as a comparison of the experimental and numerical operating points, to confirm the validity of the numerical model. What do the authors mean when referring to the flow field entropy structures? What is an entropy structure?

A: As suggested by your comment, the paper has been added more descriptions of the flow including the images. The comparison of the flow field structure in different operating points is also made during observation. Because of the restriction of authority, the operating data for the current compressor is limited. Based on existing data described in the manuscript, the fault features of the blade vibration are well captured. To a certain extent, the validity of the numerical model is checked by the comparison with the experiment. Within the scope of the author's understanding, a high entropy in filed means the flow loss, which is caused by the separation vortices. So an entropy structure here refers to the structure of the flow loss, especially the vortices leading to the blade vibration. And to capture the factors is the aim of this work.

Q: Descriptions of numerical approach: Include a more detailed description of the compressor rig. E.g. Blade counts, RPM, operating point.

A: A detailed description of the compressor has been made in the new version, including the number of the blade, rotating speed.

Q: Descriptions of numerical approach: Alongside the total number of cells used in the mesh please provide details of their distribution across the blade rows and state the radial resolution. How many time-steps were used to resolve the flow unsteadiness? What y+ values does the mesh. Is the mesh low Reynolds or are wall functions being used? Why was the strut included in the numerical model? Was this important? 

A: The distributions of the cells in the used mesh are detailed provided in the Table 1 with the circumferential、axial、radial direction resolution. Y-plus of the meshes near the blade surface is kept smaller than 10 to comply with the wall function constraint. The standard k-ε function is used as the turbulence model for the wall function by dealing with a coarse grid. During the unsteady computations, every rotor blade pitch is divided into 20 time steps with each time-step including a maximum of 20 inner interaction steps. Then the time step is calculated to be 1.54e-5s. Convergent numerical results are obtained at least after 5 revolutions. As the strut being existence in the multistage compressor, in order to avoid ignoring induction factors the struts are included in the numerical model. But from the view of current results, the struts seem not to be a direct contact with the blade vibration. 

Q: Descriptions of numerical approach, Figure 3: I don’t see the necessity of this figure and suggest it be dropped.

A: The Figure 3 is deleted. 

Q:Descriptions of numerical approach, Figure 4: As all blades are meshed identically, it should be sufficient to simply display the mesh of a single blade. Although, I think this figure could be dropped as it has no real direct relevance to the study.

A: The Figure 4 is replaced with a single finite element blade model.

Q:Numerical reconstruction of…, Figure 5: Again, it should be sufficient to show the displacement (eigenmode) of a single blade. 

A: The Figure 5 is also replaced with a single blade. 

Q:The Flow Entropy of Structure of NSV: To aid understanding the complex flow physics I strongly suggest including time-sequences to help the reader better understand your observation. Presumably, this would also make it easier to describe the observed phenomena. 

A: As the suggestion the paper has been added more descriptions about the flow, including the performance at different operating points and flow analysis of the time-sequences. The results show that the disturbances of the spiral vortexes propagate at 54.5% of the rotor speed, indicating a pattern as modal oscillation. When the frequency of blade sweeping vortex coincides with the natural mode of the blade, the blade can be emerged a large amplitude vibration. And the frequency to sweep the vortexes by the blade becomes the frequency of blade asynchronous vibration.

The typographical and grammatical errors had been revised most in the revised paper. We have made the correction accordingly. Thanks again for your advice.

Reviewer 2 Report

Investigation on the Flow Field Entropy Structure of Non-Synchronous Blade Vibration in an Axial Compressor

The Authors: Mingming Zhang and Anping Hou

Reviewer comments:

1) Title, keywords and throughout the paper text – word “compressor” should be replaced with “turbocompressor”. In the paper is not investigated standard compressor which increases operating medium pressure by using the periodical change of operating area volume – it is investigated axial high speed turbocompressor which increases operating medium pressure during its flow through the rotor and stator vanes. As in turbocompressors did not occur periodical change of operating area volume – the more correct term is a turbocompressor instead of only compressor.

2) In the Abstract should be added the most important results obtained during the analysis (their values). The description of such results is correctly presented in the Abstract, but at least some of the obtained values should be added.

3) Figure 4 should be enlarged for better visibility.

4) In Section 2 is missing analysis and presentation of results about: Grid density impact on numerical solution and Time step definition for the numerical calculation. These elements are common in the presentation of CFD simulation results, so it should be performed and involved in this section.

5) In Section 3, in my opinion, is not presented a proper validation. It is only stated that the obtained results are in agreement with the experiments from the literature. In this section should be added a diagrams and/or tables which presents a direct comparison of validation values (for various variables) obtained by simulation and obtained by measurements. Such direct comparison is completely missing and should be added in the paper, because only a statement is not sufficient. Also, it should be clearly stated what are acceptable deviation ranges of simulation results (in comparison to the measurements) for each validation variable.

6) In the paper should be discussed can the same conclusions and the same trend of the obtained results be expected in each stage of multistage turbocompressor because in this analysis is observed only one (the first) turbocompressor stage. Such discussion is essential because in real exploitation conditions, axial turbocompressors are multi-stage devices (due to low increase in pressure for each stage).

7) The Authors should better highlight in a paper what is scientific novelty of performed research and obtained conclusions. According to the currently presented elements, many of them can be found in the other literature. These which represent a novelty should be much better highlighted (and described, if necessary).

8) At the end of a Conclusions section should be added a guideline for a future research about this topic.

9) The References – firstly, the References are not prepared according to Entropy instructions for the Authors (fonts, bold and italic parts, etc.). Also, the Authors uses References dominantly older than 5 or 10 years. According to used References, it seems that this research field is not so important nowadays and that this paper is not relevant at the moment. I cannot agree with this statement, so the Authors must include in the References list a newer scientific literature (not older than 2-3 years) to confirm the relevancy of performed research.

Author Response

Reply to Reviewer

Paper: entropy-1006425

Title: Investigation on the Flow Field Entropy Structure of Non-Synchronous Blade Vibration in an Axial Compressor

Thank you very much for the suggestions for our draft. We have replied point by point to your comments, along with a clear indication of red color at the location of the revised paper.

The comments and replies can be summarized as follows:

  • Q: 1) Title, keywords and throughout the paper text – word “compressor” should be replaced with “turbocompressor”. In the paper is not investigated standard compressor which increases operating medium pressure by using the periodical change of operating area volume – it is investigated axial high speed turbocompressor which increases operating medium pressure during its flow through the rotor and stator vanes. As in turbocompressors did not occur periodical change of operating area volume – the more correct term is a turbocompressor instead of only compressor.
  • A: As suggested by your comment, the word “compressor” is replaced with “turbocompressor”.
  • Q: 2) In the Abstract should be added the most important results obtained during the analysis (their values). The description of such results is correctly presented in the Abstract, but at least some of the obtained values should be added.
  • A: According to the new version, the Abstract is revised with some important values added. And the deviation of IGV is added in the key word.
  • Q: 3) Figure 4 should be enlarged for better visibility.
  • A: The Figure 4 is replaced with a single finite element blade model used.
  • Q: 4) In Section 2 is missing analysis and presentation of results about: Grid density impact on numerical solution and Time step definition for the numerical calculation. These elements are common in the presentation of CFD simulation results, so it should be performed and involved in this section.
  • A: The missing part for the grid has been provided as shown in Figure 3. The distributions of the cells in the used mesh are detailed provided in the Table 1 with the circumferential、axial、radial direction resolution. During the unsteady computations, every rotor blade pitch is divided into 20 time steps with each time-step including a maximum of 20 inner interaction steps. Then the time step is calculated to be 1.54e-5s. Convergent numerical results are obtained at least after 5 revolutions.
  • Q:5) In Section 3, in my opinion, is not presented a proper validation. It is only stated that the obtained results are in agreement with the experiments from the literature. In this section should be added a diagrams and/or tables which presents a direct comparison of validation values (for various variables) obtained by simulation and obtained by measurements. Such direct comparison is completely missing and should be added in the paper, because only a statement is not sufficient. Also, it should be clearly stated what are acceptable deviation ranges of simulation results (in comparison to the measurements) for each validation variable.
  • A: AS the comment, the comparisons between the simulation results and measurements are listed in the Table 2. For each validation variable it is pointed out whether the result is acceptable or not in the deviation range. The prediction of the vibration frequency by computation is in reasonable agreement with the experimental data, as well as the IBPA. But because of the limitation of CFD in simulating the turbulence, the characteristic frequency fAR is exceeding the range of measurement. Despite this inaccuracy, the current study demonstrates that it is sufficient to predict the NSV phenomenon by capturing the resonance of flow instability and blade vibration under considering fluid-structure interaction.
  • Q:6) In the paper should be discussed can the same conclusions and the same trend of the obtained results be expected in each stage of multistage turbocompressor because in this analysis is observed only one (the first) turbocompressor stage. Such discussion is essential because in real exploitation conditions, axial turbocompressors are multi-stage devices (due to low increase in pressure for each stage).
  • A: Thank your com The spiral vortex observed in the flow instability of blade tip is considered as the one of the main causes to the non-synchronous vibration for the present turbocompressor. For a multistage turbocompressor, there are much differences in the flow structure of each stage. And it is indeed that the non-synchronous blade vibration is observed to appear in the first stage due to a deviation of 2° in the adjustment of the adjustable guide vane in the experiment. The effect of the deviation of IGV is gradually weakened downstream in the multistage turbocompressor. Generally speaking, the flow separation of high pressure stages is much more serious because of the reverse pressure gradient. So the instability of rotating stall may be more likely evolved instead of the spiral vortex appearance described above. So whether the conclusions for the cause of NSV is applicable depends on the actual situation of each stage in the multistage turbocompressor.
  • Q:7) The Authors should better highlight in a paper what is scientific novelty of performed research and obtained conclusions. According to the currently presented elements, many of them can be found in the other literature. These which represent a novelty should be much better highlighted (and described, if necessary).
  • A: As your comment, a discussion part has been added in the new version. The differences between the studied spiral vortex, separation vortices and rotating stall cell are discussed first. Then the novelty of this paper is highlighted as an example of blade non-synchronous vibration caused by the deviation adjustment of IGV. This is rarely involved in the previous literatures. And this work needs to be continued in the following research.
  • Q:8) At the end of a Conclusions section should be added a guideline for a future research about this topic.
  • A: The Conclusions are revised, and a future work is suggested for investigation on the influence of guide vane regulation of the blade vibration in the end.
  • Q: 9) The References – firstly, the References are not prepared according to Entropy instructions for the Authors (fonts, bold and italic parts, etc.). Also, the Authors uses References dominantly older than 5 or 10 years. According to used References, it seems that this research field is not so important nowadays and that this paper is not relevant at the moment. I cannot agree with this statement, so the Authors must include in the References list a newer scientific literature (not older than 2-3 years) to confirm the relevancy of performed research.
  • A: The new literatures have been added in the manuscript to show the focus and importance of research. At last the proportion of the literatures in the past six years is accounted for more than 40% in all used references. In the introduction the descriptions of the earlier literatures have been appropriately deleted to highlight the citation of the newer literatures.

The typographical and grammatical errors had been revised most in the revised paper. We have made the correction accordingly.

Thanks again for your advice.

Round 2

Reviewer 2 Report

The Authors have performed all the required corrections. An additional explanations were added. The literature is updated and all the required elements now existing in the paper. The additional explanations in the answers on the reviewer comments were very helpful. I have no more concerns about this paper and, in my opinion, it should be accepted.